# Multitask Boosting for Survival Analysis with Competing Risks

**Alexis Bellot**

University of Oxford

Oxford, United Kingdom

alexis.bellot@eng.ox.ac.uk

**Mihaela van der Schaar**

University of Oxford and The Alan Turing Institute

London, United Kingdom

mschaar@turing.ac.uk

## Abstract

The co-occurrence of multiple diseases among the general population is an important problem as those patients have more risk of complications and represent a large share of health care expenditure. Learning to predict time-to-event probabilities for these patients is a challenging problem because the risks of events are correlated (there are competing risks) with often only few patients experiencing individual events of interest, and of those only a fraction are actually observed in the data. We introduce in this paper a survival model with the flexibility to leverage a common representation of related events that is designed to correct for the strong imbalance in observed outcomes. The procedure is sequential: outcome-specific survival distributions form the components of nonparametric multivariate estimators which we combine into an ensemble in such a way as to ensure accurate predictions on all outcome types simultaneously. Our algorithm is general and represents the first boosting-like method for time-to-event data with multiple outcomes. We demonstrate the performance of our algorithm on synthetic and real data.

## 1 Introduction

There is now significant evidence that the progressions of many diseases interact with one another such that the prediction of events of interest, for example death due to breast cancer in a population of women, will be influenced by their simultaneous risks of developing related diseases, such as cardiovascular or pulmonary diseases [19, 20]. A central problem in survival analysis is to predict the relationship between variables and survival, which is especially challenging when a number of different correlated events might occur - i.e., there are *competing risks*. Current approaches jointly model competing risks in an attempt to capture shared latent biological traits or common risk factors. In the presence of multiple events however, jointly modelling these conditions leads to predictive models that neglect individual diseases with lower incidence. Clinical prognosis tools may result in high overall accurate predictions rates while also having unacceptably low performance with respect to an underrepresented disease outcome, which strongly reduces their explanatory power for practical purposes. The design of therapies and medical planning relies on survival estimates of predictive models. Examples of similar settings can be found in many fields beyond medicine including failure analysis in engineering and prediction of multiple economic events in economics.

The focus of this work is to provide a new interpretation of boosting algorithms [11] in a multitask learning framework [8] that extends this family to *time-to-event data with multiple competing outcomes*. Motivated by the ideas discussed above, we specifically intend to leverage the heterogeneity present in large modern data sets, the complexity in underlying relationships between events/tasks and the strong imbalance often observed between events/tasks. The aim is a flexible simultaneous description of the likelihood of different events over time that we achieve by estimating *full probability distributions*, in contrast to single time prediction problems such as regression or classification.

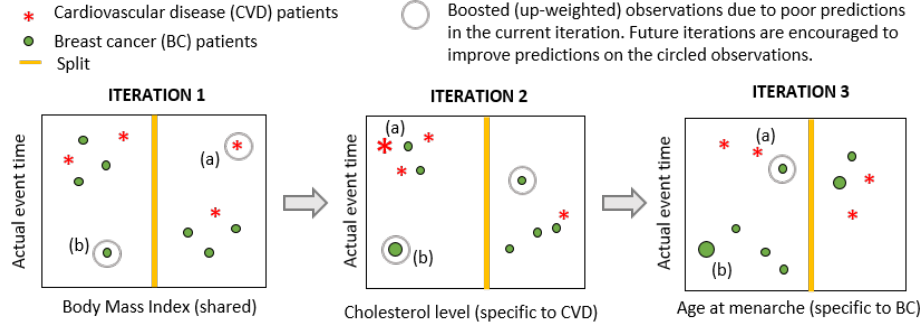

Figure 1: Boosting in the Toy Example.

We develop a boosting algorithm in which each task-specific time-to-event distribution is a component of a multi-output function. A distinctive feature is that each weak estimator (whose performance is sub-optimal) learns a shared representation between tasks by recursively partitioning the observed data (analogous to the construction of trees) from all related tasks using a measure of similarity between instances that involves *all* related tasks. This means that we learn a shared representation directly from selecting appropriate subsets of patients, which may experience different events, but which have a common time-to-event trajectory. Since this partitioning scheme is applied recursively, learned relationships (and predictions) are adaptive to the complexity of the problem and, in addition, no assumptions on the data generating process, such as accelerated failure times or proportional hazards (common in most survival models [22, 17]) have to be posed. We construct an ensemble by weighting the sample data such as to bias the next weak multivariate estimator towards mis-predicted instances. What distinguishes our weighting scheme from existing boosting methods is that while the output of each weak estimator is a multivariate probability distribution, the data only provides the specific event that occurred and the time of occurrence and thus we introduce new notions of "prediction correctness" that apply in our setting.

**Why is boosting useful for competing risks?**    A toy example we show in Figure 1 may help to illustrate our method. We consider a population that experiences one of two events, death due to cardiovascular diseases (CVD) or breast cancer. (For simplicity we ignore censoring). Each patient is characterized by its body mass index (BMI), cholesterol level and age at menarche. The medical fact is that increased BMI increases the risk of both breast cancer and CVD; increased cholesterol increases the risk of CVD but is irrelevant for breast cancer; increased age at menarche decreases the risk of breast cancer but is irrelevant for CVD. (Note that the same patients are represented in all panels of Figure 1 - the vertical position remains the same while their horizontal position changes due to different features being considered). The panels show three iterations of boosting using a stump as a weak predictor; the best partition of the data in each case is shown with the yellow threshold. The first stump recognizes BMI as best separating event times (on average), but mispredicts survival of patient (a) (that has high survival despite having high BMI) and (b) for which the contrary is true. Iteration 2, encouraged by the higher weight of (a), considers a split along the cholesterol level and is able to better describe (a)'s survival (its high survival is due to a low cholesterol level). Iteration 3, after repeatedly mispredicting (b) in iteration 1 and 2, splits based on age at menarche which explains (b)'s low survival.

Survival data in the presence of competing risks is often highly heterogeneous, the process of boosting is effective for identifying patients that do not conform to a general pattern; a fact further exacerbated when only few examples are available from each type or if imbalance is large.

## 2   Problem Formulation

The set up we consider is best defined within the context of medical patients at risk of mutually exclusive outcomes, such as causes of death, referred more generally as tasks in other domains. In this context the goal of multitask learning is to estimate cumulative incidence functions (CIFs):

$$F_1, ..., F_K : \mathcal{X} \times \mathcal{T} \rightarrow [0, 1] \tag{1}$$

$F_k$ represents the probability of a specific event of type $k$ happening before time $t$, $F_k(t|X) = p(T \leq t, Z = k|X)$[1]. This relationship is estimated from an observational sample of the random tuple $\{\boldsymbol{X}, T, Z\}$ where the input space $\mathcal{X}$ describes patient characteristics - typically $\mathbb{R}^d$ -, $T \in \mathbb{R}^+$ defines the time to event and $Z$ is the type of event observed $Z \in \{\emptyset, 1, ..., K\}$. A particularity of time-to-event data is that often the outcome will not be observed for every patient (e.g. a patient follow-up might be interrupted) however event-free survival is known *up* to a censoring time independent of $(\boldsymbol{X}, T)$ (a common assumption in the survival literature that ensures consistency of our estimates). This is the defining property of survival data and makes our setting distinct from classical supervised problems. We write $z_i = \emptyset$ for a right censored observation and $z_i = 1, ..., K$ to denote the occurrence of one of $K$ competing events.

The key idea is to exploit the shared structure of $F_1, ..., F_K$ by estimating them jointly, rather than independently, in the hope of improving prediction performance for *all* tasks. We aim to learn prognostic models $\hat{F}_k$ such as to minimize the discrepancy between predicted and actual survival status,

$$L_k(\hat{F}) := \mathbb{E}\frac{1}{\tau}\int_0^\tau \left(I(T \leq t, Z = k) - \hat{F}_k(t|\boldsymbol{X})\right)^2 dt, \qquad L(\hat{F}) = \frac{1}{K}\sum_k L_k(\hat{F}) \qquad (2)$$

which is extended on the right hand side to consider multiple tasks simultaneously by averaging over $k$. In addition, we define the cause-specific hazard function, or subdistribution hazard,

$$\lambda_k(t|X) = \lim_{dt \to 0} p(t \leq T \leq t + dt, Z = k|T \geq t, X)/dt \qquad (3)$$

It represents the instantaneous risk of experiencing an end-point related to cause $k$ and indicates the rate at which mortality with respect to that cause progresses with time. The cumulative cause specific hazard is $\Lambda_k(t) = \int_0^t \lambda_k(s)ds$.

## 3   Model Description

In this section we present our main contribution: a nonparametric boosting algorithm for jointly estimating survival distributions for multiple tasks we call Survival Multitask Boosting (`SMTBoost`). Boosting algorithms iteratively train simple predictive models on weighted samples of the data such as to encourage improvement on those data points that are mis-predicted in previous iterations. The following subsections will detail first the training procedure of weak predictors and then will provide the ensemble approach that results in flexible task-specific time-to-event distributions.

### 3.1   Weak predictors

Weak predictors are trees composed of leaves and nodes. Leafs define a partition for the data and are responsible for making predictions and nodes guide examples towards appropriate leaves using binary splits based on boolean-valued rules. We seek a binary recursive partitioning scheme -rules that partition the data at each node- resulting in the greatest difference of task-specific and overall time-to-event distributions.

### 3.1.1   Splitting rule

The key in growing trees lies in the split rule used to recursively separate the population in homogeneous nodes. In the context of competing risks homogeneity of time-to-event outcomes otherwise measured with the log-rank test statistic and model deviance in single event survival settings are not applicable. We opt instead for a modified version of Gray's test statistic [14, 16] that explicitly compares CIFs $F_k$ between two populations. Gray proposed a non-parametric log-rank test statistic defined as a weighted sum of differences in estimates of the sub-distribution hazards $\lambda_k$, that effectively generalizes the log-rank test to competing risks. In order to measure similarity with respect to all causes simultaneously, we combine the task-specific splitting rules across the event types $k$ and optimize for a weighted sum of Gray's statistic over all tasks (weighted by an asymptotic estimate of

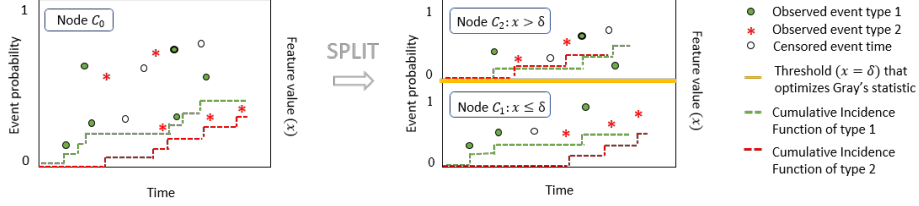

Figure 2: An example of partition of node $C_0$ into $C_1$ and $C_2$ based on Gray's test statistic for two tasks. The estimated CIFs get updated based on the resulting partition (defined by $x = \delta$) that maximizes Gray's test statistic, a composite measure of the difference between the CIFs in each subset.

the variance of each statistic, see [14] for details). Figure 2 illustrates this procedure for a single split. For conventional survival analysis, when a single event is analyzed in isolation, Gray's test statistic reduces to a log-rank test statistic commonly used in survival trees.

### 3.1.2 Leaf node predictions

Leaf nodes are responsible for predictions. Based on the final partition we compute task specific distributions with nonparametric estimators from the theory of counting processes. Let $\mathcal{C}_j$ denote the index set of training examples with leaf node membership $j$. Let $N_k(t)$ and $N(t)$ denote the number of events of type $k$ and of any type recorded before time $t$ respectively, $Y(t)$ the number of individuals at risk at time $t$ and $t_{(0)} < t_{(1)} < ... < t_{(m)}$ the ordered distinct event times (these quantities refer to leaf node $j$ only, we omit the subscript $j$ for readability). We compute task-specific survival at leaf node $j$ with the nonparametric Aalen-Johansen estimator [1],

$$\hat{F}_k(\tau) = \int_0^\tau \hat{S}(t)d\hat{\Lambda}_k(t) \tag{4}$$

where probability of event-free survival $\hat{S}$ is estimated with the Kaplan Meier estimator and the cumulative hazard function $\hat{\Lambda}_k$ is estimated with the Nelson-Aalen estimator,

$$\hat{S}(t) = \prod_{i:t_{(i)} \leq t} \left(1 - \frac{N(t_{(i)}) - N(t_{(i-1)})}{Y(t_{(i)})}\right), \qquad \hat{\Lambda}_k(t) = \sum_{i:t_{(i)} < t} \frac{N_k(t_{(i)}) - N_k(t_{(i-1)})}{Y(t_{(i)})} \tag{5}$$

The leaf nodes partition the sample space so the above construction defines the task-specific and overall cumulative incidence functions for the tree,

$$\hat{F}_k(t; \boldsymbol{x}_i) = \sum_j I(i \in \mathcal{C}_j)\hat{F}_{k,j}(t), \qquad \hat{F}(t; \boldsymbol{x}_i) = \sum_k \hat{F}_k(t; \boldsymbol{x}_i) \tag{6}$$

where the subscript $j$ refers to CIFs estimated based on each leaf of the resulting tree. This process allows to obtain completely nonparametric estimates of survival. In complex problems, and particularly from a medical perspective, this is important because subtle signals in heterogeneous populations are often unknown a priori and need to be discovered from relationships in the data.

### 3.2 Ensemble approach by boosting

In the traditional boosting framework, misclassified examples are up-weighted to bias the next weak predictor to improve previous predictions. The contrast with time-to-event settings is that model outputs are probability distributions over time and hence notions of correctness of model predictions need to be accommodated. We use the extended loss function introduced in (2), that captures the joint performance over all tasks and gives a measure of the individual empirical error:

$$e_i = \frac{1}{K\tau} \sum_k \int_0^\tau \hat{W}_i(t) \left(I(T_i \leq t, Z_i = k) - \hat{F}(t; \boldsymbol{x}_i)\right)^2 dt \tag{7}$$

where $\hat{W}_i(t)$ are estimated inverse probability of censoring weights.

Each iteration $m$ of the boosting procedure grows a tree $\boldsymbol{F}^{(m)}$ on a weighted random fraction $s$ of the training data. A decomposition of the error suggests improved performance with uncorrelated weak predictors (see the supplementary material), which we encourage by randomly sub-sampling the data; empirically we have found this to impact performance favourably. The algorithm proceeds by re-weighting all samples in our training data as a function of the individual prediction error $e_i^{(m)}$ with (7) and a measure of overall confidence in predictions of the $m^{th}$ tree, $\beta^{(m)}$. Those examples with poor event predictions get increased updated weights where $\beta^{(m)}$ is set to control the magnitude of this update: more confident trees leading to larger updates. $\beta^{(m)}$ is adjusted to lie in the interval $(0, 1)$; note that random guessing of survival probabilities results in an average error $\epsilon^{(m)}$ of $1/3$ which any weak predictor $\boldsymbol{F}^{(m)}$ is assumed to improve upon. Final time-to-event distributions for each task are computed from a weighted average of all weak predictors, weighted proportional to their confidence $\beta^{(m)}$. In contrast to existing boosting methods [9, 25, 11, 13], the output is not in the form of a discrete class label or a real valued number, but a set of distribution functions. One of the main contributions of this paper is to explicitly extend discrete label boosting to nonparametric multivariate density estimation. We present all algorithmic details in Algorithm 1.

---

**Algorithm 1** Multitask Boosting

**Input:** time-to-event data with multiple tasks $\mathcal{D} = \{(X_i, T_i, Z_i)\}_i$ of size $n$, number of iterations $M$, initial weights $w_i^{(1)} \propto 1$, sampling fraction $s$.

**for** $m = 1$ **to** $M$ **do**

    1. Let $\mathcal{D}^*$ be a randomly sampled fraction $s$ of training data $\mathcal{D}$ with distribution $w^{(m)}$.

    2. Learn weak model $\boldsymbol{F}^{(m)} : \mathcal{X} \times T \to [0, 1]^K$ on $\mathcal{D}^*$.

    3. Calculate prediction error $e_i^{(m)}$ for each instance $i$ with equation (7).

    4. Calculate adjusted error of $\boldsymbol{F}^{(m)}$, $\epsilon^{(m)} = \sum_i e_i^{(m)} w_i^{(m)}$.

    5. Calculate confidence in individual weak models $\beta^{(m)} = \frac{\epsilon^{(m)}}{2/3 - \epsilon^{(m)}}$:

    6. Update data distribution $w_i^{(m+1)} \propto w_i^{(m)} (\beta^{(m)})^{1 - e_i^{(m)}}$.

**end for**

**Output:** Final predictions $\boldsymbol{F}_f$, the weighted average of $\boldsymbol{F}^{(m)}$ for $1 \le m \le M$ using $\log(1/\beta^{(m)})$ as the weight of model $\boldsymbol{F}^{(m)}$.

---

### 3.3 Variable importance

Understanding the influence of variables on each specific task is of crucial importance in medicine and other domains. The approach we use is based on a comparison of the prediction error (2) when a variable is randomly shuffled (such that the dependence between the response and the variable in broken) in comparison to the original best fit, similarly to [26] who have shown similar procedures to be effective in many practical settings. The randomization effectively voids the effect of a variable. The intuition is that variables used as splitting rules in many tree configurations will significantly alter individual predictions (when the variable value is shuffled in each patient) suggesting high predictive power relative to other variables. Let $e_{m,j}^*$ denote the error of tree $m$ over the training data with variable $j$ randomly shuffled and $e_m$ the error without shuffling. We define the importance of variable $j$, as the weighted average of prediction error differences,

$$\frac{\sum_m \log(1/\beta^{(m)})|e_m - e_{j,m}^*|}{\sum_m \log(1/\beta^{(m)})} \tag{8}$$

Task-specific variable importance measures can be computed by considering the error only over the task specific component (i.e. using $L_k$ instead of $L$ in equation 2).

## 4 Related Work

Survival analysis under competing risks departs from more common supervised learning problems by asking both *what event will occur* and *when that event will occur*. A number of recent papers [22, 17, 5, 3, 4] only consider a single event of interest and are thus not directly applicable to

our context. We focus instead on contrasting with approaches which, like the present paper, treat competing risks.

**Parametric models** The most common techniques for the analysis of such data model explicitly some form of a cause-specific hazard (presented in (3)) as a parametric function of descriptive variables. [21] and later [10] are familiar examples of this approach. Applications of boosting [23, 15, 6], albeit in a gradient boosting framework which is very different to ours, have been proposed to improve parameter optimization by pursuing parameter updates that result in the steepest ascent of Cox's partial likelihood. With the exception of [6] all other works only consider one event types. A major downside of all the above is their dependence on proportional hazards - hazard rates between any two patients need to be in constant proportion over time - and their need to specify covariate interactions beforehand. By contrast, our work makes no such assumptions.

**Tree-based models** Closer to our work are tree-based approaches to competing risks. [16] extended Random Forests [7] to time to event estimation under competing risks. They propose a parallel ensemble in which fully grown trees are built independently on a bootstrapped sample of the data. As was empirically observed in classification problems in [24], the performance on important subsets of the population is undermined by the small contribution of underrepresented tasks to the construction of each tree. For this reason several authors [12] have suggested modifications that re-balance the data by over/under sampling subsets of the data. However our experiments using this approach, reported in section 5 produced only mixed results. By boosting, our model implicitly corrects for this imbalance by encouraging successive trees to improve performance on underrepresented tasks when they are mis-predicted. [5] use multivariate random forests within a parametric Bayesian mixture model in which the components of the mixture describe each task individually.

**Other Machine Learning models** The approach to competing risks in terms of a multi-tasking learning problem is not new to the present work. For example, [2] builds a model that couples this point of view with a representation in terms of deep multi-task Gaussian Processes with vector-valued kernels. However, the objective in [2] is to predict fixed time risk (e.g. 1 year mortality) rather than to predict full survival curves, which is our objective here. [18] is closer to the present work in that it shares the objective of predicting cause-specific survival probabilities, but the methodology is quite different, exploiting a deep learning architecture with shared and task specific layers.

# 5 Experiments

## 5.1 Evaluation Protocol

We measure performance with a common metric used in the literature: the cause-specific concordance index ($C$-index). Formally, we define the (time-dependent) concordance index ($C$-index) for a cause $k$ as follows [27]:

$$C_k(t) := \mathbb{P}(\hat{F}_k(t; X_i) > \hat{F}_k(t; X_j) | \{z_i = k\} \wedge \{T_i \leq t\} \wedge \{T_i < T_j \vee \delta_j \neq k\}) \qquad (9)$$

where $\hat{F}_k(t; X_i)$ is the predicted CIF for a test patient $i$. The time-dependent $C$-index as defined above corresponds to the probability that predicted cause-specific survival probabilities are ranked in *accordance* to the actual observed survival times given the occurrence of an event and corresponding cause. The $C$-index thus serves as a measure of the discriminative power for a cause of interest of a model and can be interpreted as an extension of the AUROC for censored data. Random guessing corresponds to a $C$-index of $0.5$ and perfect prediction to a $C$-index of $1$.

**Baseline Algorithms** We compare our model with 9 baseline algorithms described in section 4. We consider the proportional hazards model on the cause specific hazard (Cox) [21], the proportional hazards model on the subdistribution hazard (Fine-Gray) by [10] and the boosting approach to parameter optimization from [6]. These three baselines encode a linear effect of variables on survival and do not require hyper-parameter tuning except [6] for which the number of boosting iterations is optimized by cross-validation. As nonparametric alternatives we consider Random Forests for survival data under competing risks (RSF) [16] and also a weighted version (Weighted RSF) that attempts to mitigate task imbalance by sampling low incidence instances with higher probability such as to achieve balanced tasks in each bootstrapped sample. The size of the ensembles was optimized by cross-validation while the remaining hyper-parameters were set to default values. We compare with the Gaussian process model (DMGP) [2] with the suggested hyperparameters configurations and the

Deep Learning architecture (DeepHit) [18] with hyperparameters optimized with a validation set. We have in addition evaluated SMTBoost on each cause separately, denoted SMTBoost (sep.), by using the logrank test statistic instead of Gray's test (see section 3.1.1) and the deep neural network for survival prediction (DeepSurv) [17] - also evaluated on each cause separately as it does not consider competing risks - to understand the benefit of considering all causes jointly. On all experiments we train `SMTBoost` with a tree-depth of 3 and 250 boosting iterations, our default parameter settings.

## 5.2 Synthetic Studies

This section explores the ability of `SMTBoost` to recover complex survival patterns.

### 5.2.1 High Dimensional and Heterogeneous data

$$\boldsymbol{X} \sim \mathcal{U}(0,1), \quad T^1 \sim \exp(X_1^2 + \sin(X_2 + X_3) + 2X_4 + 2X_5), \quad T^2 \sim \exp(X_1 + X_2 + X_3 + 2X_6 + 2X_7)$$

This challenging setting mimics data that might be expected in genetic studies or medical data from electronic health records in which the two tasks reflect heterogeneous interactions between patient variables. We generate $1000$ event times $T$, each with probability $0.5$ from tasks 1 or 2, based on 100 variables drawn from a uniform distribution. A random subset of $20\%$ of generated times are censored by transforming their event time: $C \leftarrow \mathcal{U}(0,T)$. Only a very small number of variables, 7 out of 100, are set to influence time to event. The first 3 generated variables are shared between tasks, while variables 4 and 5 influence task 1 only, and variables 6 and 7 influence task 2 only. *All* remaining variables are introduced as noise.

As a first experiment we aim to evaluate and demonstrate the validity of our task specific variable importance procedure introduced in section 3.3. Results (normalized) are shown in Figure 3. For each variable two estimates are presented: one deriving from the error on task 1 predictions only, and one considering the error on task 2 only. We note first that even in high dimensional settings with a lot of noise, `SMTBoost` is able to distinguish between influential and noise variables. In addition `SMTBoost` captures the larger effect of task-specific variables but, due to the presence of censoring, also overestimates the importance of variables that are present in only one of the two tasks.

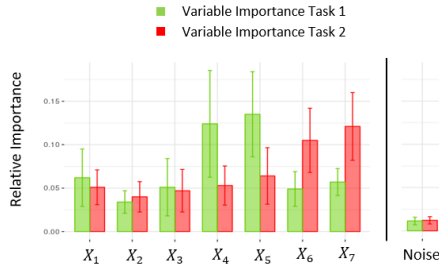

Figure 3: Variable importance in high dimensional setting.

In a second experiment we evaluate model task-specific predictions in comparison to baseline algorithms introduced in section 5.1 with the $C$-index averaged over all time horizons $t$. We present these results on the two last columns of Table 1. Performance on task 1 demonstrates the representation ability of the more flexible approaches (tree based, deep learning and gaussian process) but outperform only marginally for task 2 which has linear covariate influence. The performance of the tree-based approaches on both tasks suggests that these are more efficient in high-dimensional settings. In comparison to `SMTBoost` we believe that it is the stronger focus (by boosting) on divergent instances that leads to the gain in performance with respect to RSF and weighted RSF because the exponential transformation of covariate interactions for both tasks leads to highly divergent event times even between observations that have similar covariate values.

## 5.3 Real data studies: SEER

We investigate a patient population extracted from the Surveillance, Epidemiology, and End Results (SEER) repository similarly to [2]. The data contains $72,809$ patients of which $14.4\%$ died due to Breast cancer, $1.7\%$ due to cardiovascular diseases (CVD) and $6.1\%$ due to other causes. The remaining patients were censored. We give a more detailed description in the Supplementary material. Performance is evaluated with the cause-specific $C$-index (section 5.1), averaged over equally spaced times of 10 months from registration to the last observed event. Table 1 gives all performance results; these are averages over 4 fold cross-validation estimates and confidence bands are standard deviations.

| Models | Breast Cancer | CVD | Other | Synthetic $T^1$ | Synthetic $T^2$ |
|---|---|---|---|---|---|
| Cox | $0.773 \pm 0.02$ | $0.639 \pm 0.03$ | $\mathbf{0.688 \pm 0.02}$ | $0.612 \pm 0.01$ | $0.705 \pm 0.01$ |
| CoxBoost | $0.774 \pm 0.02$ | $0.642 \pm 0.03$ | $0.678 \pm 0.02$ | $0.613 \pm 0.01$ | $0.705 \pm 0.01$ |
| Fine-Gray | $0.777 \pm 0.02$ | $0.636 \pm 0.03$ | $0.682 \pm 0.02$ | $0.613 \pm 0.01$ | $0.706 \pm 0.01$ |
| RSF | $0.789 \pm 0.03$ | $0.722 \pm 0.03$ | $0.643 \pm 0.02$ | $0.654 \pm 0.01$ | $0.720 \pm 0.01$ |
| Weighted RSF | $0.778 \pm 0.03$ | $0.730 \pm 0.03$ | $0.645 \pm 0.02$ | $0.645 \pm 0.01$ | $0.717 \pm 0.01$ |
| DeepHit [18] | $0.800 \pm 0.01$ | $0.662 \pm 0.01$ | $0.684 \pm 0.01$ | $0.652 \pm 0.01$ | $0.720 \pm 0.01$ |
| DMGP [2] | $0.801 \pm 0.02$ | $0.732 \pm 0.03$ | $0.646 \pm 0.02$ | $0.651 \pm 0.01$ | $0.718 \pm 0.01$ |
| DeepSurv [17] | $0.781 \pm 0.02$ | $0.659 \pm 0.03$ | $0.685 \pm 0.03$ | $0.629 \pm 0.01$ | $0.710 \pm 0.01$ |
| SMTBoost (sep.) | $0.795 \pm 0.02$ | $0.721 \pm 0.04$ | $0.660 \pm 0.03$ | $0.631 \pm 0.01$ | $0.710 \pm 0.01$ |
| SMTBoost | $\mathbf{0.819 \pm 0.02}$ | $\mathbf{0.766 \pm 0.03}$ | $\mathbf{0.688 \pm 0.02}$ | $\mathbf{0.664 \pm 0.01}$ | $\mathbf{0.721 \pm 0.01}$ |

Table 1: $C$-index figures (higher better) and standard deviations on the SEER and synthetic dataset.

**Source of gain** Patients suffering from chronic diseases tend to be very heterogeneous, mortality rates can be highly divergent even within narrow phenotypes. The limitations imposed by proportional hazard models to model this kind of data are evident from the performance results on both Breast Cancer and CVD outcomes. Predictions of other causes tend to benefit from simpler modelling approaches as SEER predominantly records patient information related to Cancer (see Supplement) which suggests that few predictive variables are available for other causes. Performance gains of SMTBoost are largest with respect CVD outcomes which illustrates its ability to handle low incidence tasks (only $1.7\%$ of events relate to CVD). Both DeepHit and DMGP are competitive as they leverage the influence of shared risk factors but underperform SMTBoost. The results suggest that boosting to handle imbalance is crucial to improve predictions.

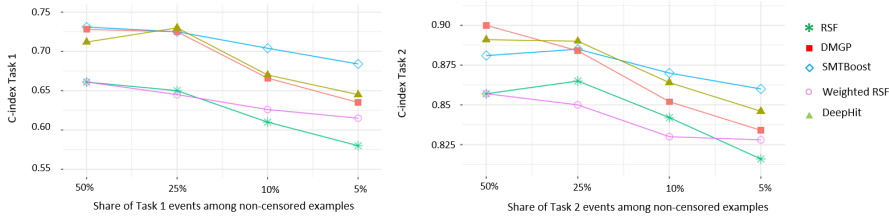

Figure 4: Mean $C$-index results (higher better).

### 5.3.1 Further exploring imbalanced heterogeneous data

We constructed an additional more general synthetic experiment designed to express complex and heterogeneous survival patterns between 2 tasks to further understand performance in imbalanced data sets. Consider the following data generation process,

$$\boldsymbol{X} \sim \mathcal{U}(0,1), \quad T^1 \sim \exp(\log(\alpha_1^T \boldsymbol{X}) + \alpha_2^T \boldsymbol{X}^2), \quad T^2 \sim \exp(\alpha_3^T \boldsymbol{X})$$

Variables $\boldsymbol{X}$ and parameters $\alpha_1, \alpha_2, \alpha_3$ are each of dimension 5 whose components are drawn at random from a uniform distribution. For each task we investigate predictive performance as a function of task prevalence by analyzing 4 scenarios with different task proportions in the resulting data. For instance a first balanced scenario for task 1 would involve a split: $50\%$ censored, $25\%$ task 1, $25\%$ task 2. We generate 5 data sets (by sampling variables and parameters randomly) of 1000 instances for each individual scenario and set a random $50\%$ of the population to be uniformly censored. We show performance results in Figure 4, as a function of task 1 and task 2 occurrence in the data. As expected, all models have their performance deteriorate the fewer samples available but we observe increasing relative performance gains for both SMTBoost and weighted RSF, the only two approaches that attempt to correct for the imbalance.

# 6 Conclusion

We have introduced a boosting-based algorithm for survival analysis with multiple outcomes, designed to handle the heterogeneity present in modern medical data sets, including highly imbalanced data and high dimensional feature spaces. Our experiments on synthetic and real medical data have demonstrated large performance improvements over current techniques and show the advantage of a boosting framework, already observed in classification and regression problems, in the field of time-to-event analysis. From a medical perspective our model contributes towards the field of "individualized medicine". Our hope is that based on our model clinicians can improve long term prognosis and more accurately weight the benefits of a treatment for each *individual* patient whose characteristics may lead her to behave differently than the average.

## Footnotes

[1]Also called subdistribution function because it does not converge to 1 as $t \to \infty$, but to $p(Z = k)$, the expected proportion of task $k$ events. However, the CIFs for all possible event types will always add up to the distribution function of $T$.

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
