[Supplementary Material]

# Supplementary Material: Multitask Boosting for Survival Analysis with Competing Risks

**Alexis Bellot**
University of Oxford
Oxford, United Kingdom
alexis.bellot@eng.ox.ac.uk

**Mihaela van der Schaar**
University of Oxford and The Alan Turing Institute
London, United Kingdom
mschaar@turing.ac.uk

This Supplement contains additional methodological details and experiments analyzing the hyperparameter specifications of `SMTBoost`, its computational complexity and motivation for incorporating randomness through subsampling as an integral part of the training procedure. Finally we give further information on Gray's test for equality in subdistribution hazards and on the experiments conducted on the data set from SEER.

## 1 Motivation for incorporating stochasticity

We define the mean squared error as the expected integral of the squared difference between the estimated and true cause-specific cumulative incidence functions. Here $\hat{F}$ is the multivariate output of our algorithm with component $k$ the predicted CIF for cause $k$ of the ensemble of trees and $F_k$ is the true underlying cumulative incidence function for cause $k$.

$$
\begin{aligned}
MSE_k(\hat{F}, \tau) =& \mathbb{E} \int_0^\tau \left( \hat{F}_k(t; \boldsymbol{x}) - F_k(t; \boldsymbol{x}) \right)^2 dt \\
=& \mathbb{E} \int_0^\tau \left( \hat{F}_k(t; \boldsymbol{x}) - \mathbb{E}\hat{F}_k(t; \boldsymbol{x}) \right)^2 + \left( \mathbb{E}\hat{F}_k(t; \boldsymbol{x}) - F_k(t; \boldsymbol{x}) \right)^2 dt \\
=& \mathbb{E} \int_0^\tau \left( \sum_m \hat{\gamma}^{(m)}(\hat{F}_k^{(m)}(t; \boldsymbol{x}) - \mathbb{E}\hat{F}_k^{(m)}(t; \boldsymbol{x})) \right)^2 + \left( \sum_m \hat{\gamma}^{(m)}(\mathbb{E}\hat{F}_k^{(m)}(t; \boldsymbol{x}) - F_k(t; \boldsymbol{x})) \right)^2 dt \\
=& \int_0^\tau \sum_m \sum_j \hat{\gamma}^{(m)}\hat{\gamma}^{(j)} \bigg( \mathbb{E}\left( (\hat{F}_k^{(m)}(t; \boldsymbol{x}) - \mathbb{E}\hat{F}_k^{(m)}(t; \boldsymbol{x}))(\hat{F}_k^{(j)}(t; \boldsymbol{x}) - \mathbb{E}\hat{F}_k^{(j)}(t; \boldsymbol{x})) \right) + \\
& \quad \mathbb{E}\left( (\mathbb{E}\hat{F}_k^{(m)}(t; \boldsymbol{x}) - F_k(t; \boldsymbol{x}))(\mathbb{E}\hat{F}_k^{(j)}(t; \boldsymbol{x}) - F_k(t; \boldsymbol{x})) \right) \bigg) dt \\
=& \sum_m (\hat{\gamma}^{(m)})^2 MSE_k(\hat{F}^{(m)}) + \\
& \quad \mathbb{E} \sum_m \sum_{j \neq m} \hat{\gamma}^{(m)}\hat{\gamma}^{(j)} \int_0^\tau \left( (\mathbb{E}\hat{F}_k^{(m)}(t; \boldsymbol{x}) - F_k(t; \boldsymbol{x}))(\mathbb{E}\hat{F}_k^{(j)}(t; \boldsymbol{x}) - F_k(t; \boldsymbol{x})) \right) + \\
& \quad \text{Cov}(\hat{F}_k^{(m)}(t; \boldsymbol{x}), \hat{F}_k^{(j)}(t; \boldsymbol{x})) dt
\end{aligned}
$$

where $\hat{F}_k(t; \boldsymbol{x}) = \sum_m \hat{\gamma}^{(m)}\hat{F}_k^{(m)}(t; \boldsymbol{x})$. Note the fact that $F_k(t; \boldsymbol{x}) = \sum_m \hat{\gamma}^{(m)}F_k(t; \boldsymbol{x})$ since $\sum_m \hat{\gamma}^{(m)} = 1$, used in line 3. Hence everything else being equal, lowering the correlation between successive weak learners reduces the mean squared error. This decomposition motivates combining trees trained on different samples of the data as their correlation tends to be lower.

## 1.1 Dependence on hyperparameter specification

The performance of the boosting architecture will likely depend on the amount of randomization introduced and complexity of the ensemble. In this section we investigate the impact of randomness and the impact of the complexity of the ensemble on performance. We consider the same synthetic data scenario as in section 5.2.1 in the main body of this paper. Panel (a) of Figure 1 compares the cause-specific $C$-index for task 1 (averaged over time) at different rates of subsampling. The results suggests that subsampling in the range $0.7 - 0.8$ improves performance. Panel (b) shows the impact of the number of trees on the $C$-index evaluated out-of-sample by cross-validation.

Figure 1: Hyperparameter specification impact on predictive performance of `SMTBoost`.

## 2 Computational Complexity

The computational complexity of `SMTBoost` is $\mathcal{O}(W(N, D, K)M + DNlogN))$, where $W$ represents the complexity of growing a tree as a function of $N$ the number of patients, $D$ is the number of covariates and $K$ the number of tasks. The burden of the complexity lies in the construction of the trees as the rest of the operations can be performed in $\mathcal{O}(N)$. Assuming the data samples are sorted in each variable, finding the best tree requires $\mathcal{O}(NDK)$ operations since we need to compute gray's test statistic for all $K$ tasks along all possible splits. Sorting all the covariates will take $\mathcal{O}(DNlogN)$ time and this has to be done only once before starting the first iteration. Hence, the overall cost of $M$ iterations is $\mathcal{O}(NDKM + DNlogN)$.

## 3 Gray's test for subdistribution hazards

We used Gray's test [1] as a splitting criterion in our tree construction to evaluate the null hypothesis of equality of cause-specific cumulative incidence functions between two groups, $H_0 : F_k^1(t) = F_k^2(t)$ for all $t > 0$ where 1 and 2 denote the two groups. Gray's test statistic with tractable distribution under the null is defined in terms of the hazard (defined in section 2 of the main body of this paper).

$$G := \int_0^\tau K(t)(\hat{\lambda}_j^1(t) - \hat{\lambda}_j^2(t))dt \tag{1}$$

where we define $\tau$ to be the maximum observed event time and $K$ is a weight function that specifies the importance given to earlier or later time horizons. $K$ is defined with the specification given in section 3 of [1].

## 4 Summary Statistics: SEER

The prevalence of multi-morbidity globally (for which the co-occurrence of multiple diseases has been explicitly recognized) has doubled over the last 20 years and represents now two-thirds of people aged over 65 [3]. It is increasingly important that risk prognosis be done jointly over the possible outcomes and be flexible enough to accurately model the complex relationships between them. Breast cancer and Cardiovascular diseases are two known conditions that share biological risk factors and represent the two largest contributors to the burden of chronic disease in the United States [2].

We investigate a patient population extracted from the Surveillance, Epidemiology, and End Results (SEER) Program which contains $72,156$ patients with diagnosed breast cancer but for which

cardiovascular disease related outcomes are also recorded. These patients are described by 12 covariates including: age, gender, tumor and surgery information, and a number of physiological markers related to cancer. Table 1 details the feature distribution of the extracted cohorts.

Table 1: Main feature distributions of the extracted cohort of SEER.

| Mean (Std. Dev.) | Breast Cancer | Cardiovascular disease | Other | Censored |
|---|---|---|---|---|
| # of Patients | 10462 | 1231 | 4441 | 56675 |
| Time to event (months) | 56.82 (39.9) | 58.8 (52.4) | 81.68 (46.02) | 144.9 (23.4) |
| Age (y) | 50.94 (5.8) | 38.86 (23.6) | 52.92 (5.48) | 50.82 (5.69) |
| Woman | 99.2% | 98.9% | 99% | 99.6% |
| Tumor Marker | 2.41 (10.28) | 1.24 (6.95) | 1.84 (8.77) | 0.57 (4.80) |
| Lymph nodes | 2.91 (119.27) | 0.71 (8.44) | 13.78 (839) | 1.38 (98.9) |
| Positive histology | 98.1% | 99.3% | 98.7% | 99.5% |
| Positive cytology | 1.1% | 27.7% | 0.5% | 0.02% |
| Malignant Tumors | 1.25 (0.51) | 1.18 (0.48) | 1.61 (0.74) | 1.23 (0.51) |
| Laterality (Right Breast) | 0.48% | 0.62% | 0.49% | 0.49% |
| ACJJ Stage | 5.11 (1.90) | 4.40 (2.26) | 5.49 (1.37) | 5.70 (0.89) |