[Reviews · NeurIPS 2018]

Reviewer 1



The paper tackles a timely area of research, namely new approaches for time-to-event data modeling which is common in health data analysis. There is a large body of statistics literature on this topic and some recent ML approaches, such as DeepSurvival (Blei lab) and DeepHit (van der Shaar lab). The work proposes a multi-task model to consider competing risks, similar to DeepHit. The authors do compare with other methods, but I think these comparisons fail in two aspects: a) no comparison with a version of their method that considers each risk independently -- this comparison would be important to include in order to understand whether it is the flexibility from boosting trees or from the simultaneous modeling of multiple risks that leads to the improved performance. b) a comparison with DeepSurvival (from 2016, https://arxiv.org/abs/1608.02158) appears appropriate. Apart from the limitations in the results part, I thought that the presentation of the methodology as a boosting method is not very clear. I consider myself an expert in boosting and miss a few important ingredients and explanations of the algorithm. a) what is the objective that is optimized by the proposed algorithm, b) how is the step size chosen (see Step 5 in All 1), c) is there any guarantee that the proposed algorithm converges to a minimum/maximum of the objective function. There is a lot of literature on boosting and how to these techniques are related to optimization, but there is very little mentioning of that work (except that it is very different from Gradient boosting). Without any analysis or justification beyond intuition of why the algorithm was chosen this way, it appears a bit too ad hoc. Some details: * \tau is not defined in (2) * p(T<=t, Z=k|X) is used in the definition of F_k but not really defined anywhere. I'd think that such details should be described in Section 3, but I couldn't find them there. AFTER AUTHOR RESPONSE: * The authors included a new table comparing the model against itself with separate causes. They were able to show that the results significantly improved by considering multiple causes together. (please include a statement how you determined "significance". I used a not quite appropriate t-test with the means and standard deviations that were provided with n=10 and all comparisons were significant at p<=5%). This addresses one concern I had about this work and I increase my score by one. * I still think that there is little justification for the algorithm. The optimization perspective can give justifications -- which the authors did not follow. Any other justification -- theoretical, analytical, empirical would be fine as well. As it stands the algorithm still appears ad hoc to me. So, no additional change in score here.

Reviewer 2



Edit: Post author-feedback period, the authors have addressed both the patient specific hazard as well as the variable importance well and would be interested in the final version of the manuscript. Original Review: The authors tackle the problem of modeling time-to-event datasets under competing risks. Patient health trajectories are typically affected by multiple diseases that can influence the occurrence of each other. While survival analysis has been studied in depth, most of the methods were modeled for single outcome analysis. Also, the authors note the current methods on modeling competing risks typically make certain parametric assumptions such as proportional hazards for cox models. The key idea of this paper is to adapt boosting process to model time-to-event datasets under a non parametric setting. Going a step deeper, the core contribution is to explicitly extend discrete label boosting to multivariate non parametric setting. The paper presentation is okay and while the model could have been better described, algorithm 1 helps with the overall clarity as well as reproducibility of the paper. The evaluation protocol presented in the paper is well reasoned and the results are promising. However, there are a few areas where the paper can improve: * The variable importance as described in 3.3 doesn't seem to be a natural extension of the model and can benefit from a better motivation behind the formulation * While the results are promising, only analyzing the time-dependent C-index is insufficient. For e.g. it would have been interesting to analyze particular cases with multiple comorbidity and explore how the model interpreted the same. * The paper is in a way a very clever approach by using the well studied technique of boosting and applying it in a sequential manner. However, for this to be practically useful, models need to reason behind the adjustments of hazards over time which is currently missing. A minor point about the synthetic datasets. Instead of separating out the cases for imbalanced and balanced synthetic datasets, the space could have been better utilized by analyzing the real datasets.

Reviewer 3



The authors propose a boosting method (SMTBoost) for survival analysis with competing risks. Each weak predictor is a tree that is constructed using a splitting rule based on a modified version of Gray’s test statistics. The prediction at each leaf node is given by the nonparametric Aalen-Johansen estimator. By introducing an extended loss function (eq.2), the authors extend the discrete label boosting to nonparametric multivariate density estimation, which is the core contribution of this work. While relatively straightforward, the approach performs well. Based on a series of empirical evaluations on two synthetic datasets and one real dataset, the authors show that SMTBoost outperforms all baseline methods. I've read the author response.

Reviewer 4



This paper has the aim to estimate cause-specific as well as overall survival distributions in a competing risk setting, and employs boosting algorithms in a multitask learning framework to obtain estimators. The paper is well-written and clear in its presentation, and applies established methods and algorithms to the setting. While there exist applications of all employed methods (tree-based, boosting, multitask learning) to estimation in competing-risks, which is very well described in the related work section, the paper presents novelty in the details of their approach and their goal to estimate full cause-specific and all-cause hazards. Further, evaluations on both synthetic as well as real-data show an improvement of prediction compared to existing approaches. Strengths of the paper: - Aim of the paper is clearly stated and mathematical notation is well defined. - The related work section is well-written, and their proposed approach is compared with different existing approaches. - Synthetic and real-data applications are described in a reproducible manner and show that the proposed approach can yield an improvement in prediction in multiple scenarios. - Nice idea of visualizations in Figures 1-2, which help to get an illustration of boosting and intuition on the partitions (but which could be improved, see below), further the description throughout the paper is easily accessible with very few typos. Some comments: - Description of the toy example in Figure 1 and lines 52-67 could be improved: when reading the text and looking at the figure, it was not clear to me if the goal in the example was to improve the estimation of cause-specific hazards (i.e. for death due to CVD and death due to BC separately) or of all-cause hazard by using boosting. The presentation here seems to focus on all-cause hazards while the later synthetic and real-data examples seem to focus on cause-specific hazards, which was a bit confusing to me and might be improved, e.g. in line 61 (“… best partition of the data …”): best partition with respect to what, high/low overall survival? - Line 2: “… have a higher risk …” instead of “… have more risk …”? - Line 75: add “given covariates“ in “represents the probability of a specific event of type k happening before time t given covariates”? - Line 77: What can \mathcal{X} and \mathcal{T} be, e.g. \mathbb{R}^k and \mathbb{R}^+ ? - Line 98: results in flexible estimates of task-specific time-to-event distributions? - Lines 227ff: it could help the reader to explicitly state that \bold{X} encompasses X_1 … X_100 - Lines 109-115: Here, it could be helpful to remind the reader what exactly is tested using Gray’s test statistic, i.e. if the equality of all subhazards is tested vs. that at least two subhazards differ. - Figure 2: I looked at the example in the figure to get an intuitive understanding how the partition is obtained, i.e. which partition maximizes Gray’s test statistic. As the authors describe in the figure legend, the partition is supposed to maximize “a composite measure of the difference between the CIFs in each subset”. Now by looking at the figure, I was wondering if other partitions might not yield a great difference between the CIFs, and I was confused that the green dot with third-lowest feature value is different in the left and right plot. Is this point upweighted in the right plot, or is this a typo? - In the introduction (line 25-26), the authors describe that existing approaches can do well in prediction of all-cause survival but not well for cause-specific survival. As far as I could tell, their proposed approach was only compared in its estimation of cause-specific survival to existing approaches. How does it compare in prediction of overall survival? - What is shown in Table 1 in the supplements? Mean and SD? - The description and algorithm focuses on competing risks, and it seems to me that there is potential to include recurrent events (e.g. not only focus on death due to CVD/BC but also re-occurrence of BC) in the model as future work. AFTER AUTHOR RESPONSE: I have read the author response, and think that the paper will be further strengthened by the corrections and additional results described in the rebuttal letter.